# Are Clean Technologies More Effective Than End-of-Pipe Technologies? Evidence from Chinese Manufacturing

**DOI:** 10.3390/ijerph18084012

**Published:** 2021-04-11

**Authors:** Jiawei Li, Jianghong Zeng, Zhengke Ye, Xiangrong Huang

**Affiliations:** 1Department of Management Science and Engineering, Business School, Central South University, Changsha 410083, China; 171601008@csu.edu.cn; 2Department of Marketing, School of Business Administration, Hunan University of Finance and Economics, Changsha 410083, China; huangdora@126.com

**Keywords:** environmental technologies, financial constraints, market advantages, economic performance

## Abstract

An increasing number of manufacturing enterprises are adopting environmental technologies to cope with the increasingly severe environmental regulatory pressure, but the existing studies about the impact of environmental technologies on economic performance come up with mixed results. This paper contributes to the literature by using the financial constraints and market advantages as a dual mediating process in this relationship. An empirical test using a sample of Chinese manufacturing enterprises listed in the Shanghai and Shenzhen exchange from 2011 to 2018 is established. The results of regression analyses show that end-of-pipe technologies and clean technologies have a positive effect on firms’ economic performance. Moreover, we find that clean technologies not only directly affect economic performance but also indirectly affect economic performance through mitigating financial constraints. With the negative influence of end-of-pipe technologies on market advantages, the positive economic effect caused by end-of-pipe technologies is weakened. This research provides useful insights into the selection of environmental technologies for manufacturing firms and the establishment of new policies to promote green finance and green consumption.

## 1. Introduction

An extensive economic growth pattern has caused excessive resource consumption and deterioration of the environment in developing countries, especially in China [1]. Consequently, China has implemented a series of environmental policies and emission reduction targets to achieve sustainable development [2,3]. The manufacturing sector, the largest energy consumption industry in China’s economic development, has played a primary role in driving severe air and water pollution, causing widespread health problems [4]. According to the data in the China Environmental Statistics Yearbook from 2013 to 2015, 66.14% of the total emissions of waste gas came from manufacturing, and industrial sulfur dioxide emissions from manufacturing account for 51.79% of the total volume. Meanwhile, more than 38.68% of nitrogen oxides emissions come from manufacturing. The pure pursuit of GDP (gross domestic product) at the expense of resource waste and environmental damage is bound to be unsustainable [5]. How to achieve green transformation and development has become an important direction for the future development of China’s manufacturing.

A large number of manufacturing firms in China have implemented environmental technologies as a strategic activity to reduce environmental impact [6,7]. Environmental technologies refer to the improvement of existing production processes or the addition of new processes, and are commonly divided into end-of-pipe technologies and clean technologies [8,9]. End-of-pipe technologies focus on pollution control technologies, using equipment that is added as a final process step to capture pollutants and wastes prior to their discharge. Clean technologies are defined as pollution prevention technologies, referring to fundamental changes to the manufacturing process that reduce any negative impact on the environment during material acquisition, production, or delivery.

The balance or trade-off between reducing pollution emissions and improving economic performance is key to the sustainable development of manufacturing companies. In the literature, two opposing views circulate regarding the impact of environmental technologies on firms’ economic performance. On the one hand, the traditional or cost-based view argues that under the constraints of environmental regulations, companies tend to redistribute their existing labor and capital resources to environmental technologies, which often adversely affect their economic performance [10]. On the other hand, supporters of environmental technologies hold that environmental technologies can improve the productivity and competitiveness of firms, thus increasing the profits of firms [11]. In addition, most manufacturing firms are dependent more on end-of-pipe technologies, which are conveniently added to the existing production process, ignoring the role of clean technologies [9]. Therefore, the first focus of this study is to analyze how different types of environmental technologies affect economic performance.

Previous studies have shown that corporations can obtain many benefits and resources from external stakeholders through adopting environmental technologies [12,13]. The stakeholder theory identifies the generation of value as a central driver of the enterprise, this value is to be shared by a group of stakeholders who can affect or be affected by an organization [14]. Along with green finance and green consumption spring up, investors and consumers are playing an increasing role in the profits of manufacturing enterprises. Consequently, this research further develops the second focus, that is, whether the adoption of environmental technologies responds to the requirements of stakeholders and thus improves the economic performance of the company.

There are obvious gaps in the current answers to the above questions; this statement can be explained as follows. For the first question, existing literature mainly examines the impact of environmental technologies on economic performance, while different types of environmental technologies are rarely discussed. Although Xie et al., (2016) examined the effects of green process innovation on the financial performance of manufacturing [15], they neither exploit empirical research using firm-level data nor focus on stakeholder impact. For the second question, existing literature mainly explains the motivation of firms to adopt environmental technologies from the stakeholder’s perspective. Current research on the relationship between environmental technologies and stakeholders comes up with mixed or inconclusive results [16]. It remains unclear how different types of environmental technologies affect financial constraints and market advantages.

In order to address the above gaps, this study empirically identifies the direct influence of different types of environmental technologies on economic performance at the firm level, as well as the indirect influence mechanism on financial constraints and market advantages. We use a large firm-level dynamic panel data set from 2011 to 2018 to investigate the effect of environmental technologies on manufacturing firms’ economic performance. Our findings show that end-of-pipe technologies and clean technologies have a positive effect on a firm’s economic performance. Furthermore, financial constraints and market advantages have a mediating effect on the relationship between a firm’s environmental technologies and economic performance, and its mediating effect differs depending on the types of environmental technologies. Finally, we test the endogeneity and robustness via several methods.

This paper mainly contributes to three aspects. First, prior studies concerning environmental technologies lack attention to environmental technologies. Therefore, we employ the investment in environment protection alteration and green patent as the measurement indicators in order to truly reflect technological characteristics of end-of-pipe technologies and clean technologies. Second, this paper enriches the literature related to the relationship between environmental sustainability behavior and the economic performance of manufacturing firms. It explains that environmental technologies may improve firms’ economic performance through alleviating financing constraints and market advantages, which provides new insights for manufacturing enterprises to promote green transformation. Finally, this research further supplements the stakeholder theory literature, especially regarding the investors and customers. In the context of booming green finance and green consumption, corporations can obtain more benefits and resources through clean technologies. Based on these findings, this paper provides practical guidance to manufacturing enterprises and government, enabling decisions to be made more efficiently.

## 2. Literature Review and Hypotheses

### 2.1. Environmental Technologies and Economic Performance

Regarding the relationship between environmental technologies and economic performance, the cost-based view emphasizes that costs incurred by a firm putting into environmental technologies decrease the firm’s competitiveness and economic performance [10,17,18,19]. Against that view, most scholars believe that environmental technologies had a positive impact on firms’ economic performance through production cost reduction, processes improvement, and product innovation [11,20,21,22,23]. To be specific, Chiou et al. [24] and Chan et al. [25] revealed that environmental technologies have provided firms with a competitive advantage by increasing cost efficiency and profitability. Yan et al., found that both technology- and process-based environmental innovations positively influence airline revenue [26]. Chang revealed that environmental technologies can perform corporate social responsibility, which can improve the reputation and image of firms and increase their economic performance [27].

To sum up, while the additional financial expenditure of environmental technologies, environmental technologies can improve economic performance. Economic performance can be improved with: (1) Pollution control costs decreased by reducing pollution emissions and waste recycling; (2) reducing costs at the source and increasing revenue by redesigning production processes, improving resource utilization; and (3) obtaining numerous benefits and resources from stakeholders. From the perspective of technology types, end-of-pipe technologies involve removing ex-post emissions, i.e., the by-product of manufacturing through filters, scrubbers, cyclones, or centrifuges [28]. The distinguishing feature of end-of-pipe technologies is that they do not affect the production process itself. Therefore, end-of-pipe technologies have a positive effect on economic performance by primarily reducing pollution control costs. Compared with end-of-pipe technologies, clean technologies reduce emissions ex-ante at the production level. Examples of such technologies include closed-loop production processes, switching to less polluting materials and fuels, and the replacement of coolants or encapsulation of equipment. The purpose of clean technologies is to continuously improve efficiency and minimize cost. In the long run, clean technologies are conducive to economic performance improvement because of the reductions in by-product and resource utilization. Therefore, the following hypotheses are proposed:

**Hypothesis** **1a** **(H1a):***End-of-pipe technologies are positively associated with economic performance*.

**Hypothesis** **1b** **(H1b):***Clean technologies are positively associated with economic performance*.

### 2.2. Environmental Technologies and Financial Constraints

As stakeholders are concerned more about the environment, environmental technologies may play a significant role in mitigating financing constraints. China has been implementing a green loan policy as a trial to control pollution and protect the environment through financial systems [29]. For enterprises, the impact of environmental technologies on corporate financing may be achieved through green loans. The “Green Credit Guide,” implemented in 2012 by the China Banking Regulatory Commission (CBRC) indicates that banks treat corporate environmental behavior as an important loan reference to improve the likelihood of obtaining a loan [30]. It is a well-known fact that manufacturing firms with success in environmental technologies could send a positive signal to the public. As a result, it can improve the level of the firm’s credit financing. Moreover, investors and analysts will attribute a high investment value and high future expectations to these firms committed to environmental technologies [31]. If a firm with a high level of environmental technologies issues new shares or new bonds to raise funds, it could attract more potential shareholders and creditors or increase capital from existing investors [13].

In summary, environmental technologies are an important approach to reduce financial constraints by reducing the cost of corporate credit financing, increasing external investment. Considering the different kinds of environmental technologies, end-of-pipe technologies can be easily added to the existing production processes with few barriers [32]. For banks, a loan to these firms adopting end-of-pipe technologies can simultaneously meet the requirements of the company’s solvency and the government green development. On the other hand, the main concern of investors is whether they can obtain higher future returns. The clean technologies can potentially reduce production costs in the long term, thus attracting more investment [13]. Therefore, the following hypotheses are proposed:

**Hypothesis** **2a** **(H2a):***End-of-pipe technologies are positively associated with financial constraints*.

**Hypothesis** **2b** **(H2b):***Clean technologies are positively associated with financial constraints*.

### 2.3. Environmental Technologies and Market Advantages

For enterprises, environmental technologies respond to the requirements of stakeholders and strengthen the relationship between enterprises and stakeholders. As global climate change and the deterioration of the ecological environment accelerate, the concept of green consumption, as a global trend in consumption, has been popularized [33]. Compared to traditional products, green products or services that are supported by environmental technologies can better match the values and preferences of consumers [34]. Corporations could acquire higher market advantages by proactively adopting environmental technologies activities to meet the increasing demands for green consumption [35]. Nogareda stated that eco-innovation could positively affect a firm’s competitiveness and market recognition [36]. In addition, developing environmental and social-friendly technology can obtain a first-mover advantage for novel products and create a new green market [37]. However, academics who doubt or disagree with this view are still concerned with the “unrecoverable costs” caused by green investments [19]. To breakeven on “unrecoverable costs”, suppliers may charge higher prices on green products [38,39].

End-of-pipe technologies pertain to incremental innovations because they do not affect the implementation of essential technology or modify fundamentally the production processes [9]. Therefore, customers may be unaware of the environmental benefits of end-of-pipe technologies, which are perceived as a cost burden and hamper companies’ competitiveness [40]. Related to the outcomes of environmental technologies, Ghisetti and Rennings found that clean technologies positively affect company competitiveness, while this is not the case when adopting end-of-pipeline technologies [41]. In contrast, clean technologies have the potential to reduce the use of materials and/or energy, which enhances the supplier’s brand image and reputation among customers [42]. The implementation of clean technologies satisfies the increasing demand for green consumption and gains consumer trust, with greater market advantages expected. Therefore, the following hypotheses are proposed:

**Hypothesis** **3a** **(H3a):***End-of-pipe technologies are negatively associated with market advantages*.

**Hypothesis** **3b** **(H3b):***Clean technologies are positively associated with market advantages*.

### 2.4. Mediating Role of Financial Constraints and Market Advantage

Similar to other emerging markets, enterprises in China are faced with severe financial constraints [43]. Multiple studies have examined the role of financial constraints on a firm’s performance [44,45]. In the presence of market imperfection, financial constraints could affect a firm’s investment decisions, further affecting firms’ productivity [44]. Firm survival and growth are negatively related to the difficulty faced in accessing external funds because financial constraints have a negative effect on innovation expenditures and overall investment [46]. Using firm-level data from 81 countries, Haider et al., show that firms with less financial constraint outperformed those with more financial constraint [47]. Therefore, firm economic performance improvement of manufacturing firms might be caused by mitigating financial constraints.

The literature well documents that the ultimate consequence of any competitive advantage deriving from proactive environmental management will most probably be an improvement in economic performance [48,49,50]. Firms with market advantages have successfully created a unique product to differentiate themselves from competitors and thus, can reap the benefits of high levels of customer loyalty and satisfaction [51]. On the one hand, loyal customers are less sensitive to price changes, firms can command premium prices or sell more of their products at a given price, leading to better economic performance [52]. On the other hand, the positive reputation that results from higher levels of market advantage enables the firm to attract new customers and introduce new products [19]. Therefore, market advantage has a positive effect on economic performance.

In summary, our proposed research model states that the adoption of end-of-pipe technologies and clean technologies, respectively, affects firms’ financial constraints and market advantages in different ways, and thus have different influences on economic performance.

In the case of end-of-pipe technologies, they increase the possibility that companies will receive green credit. This brings about more financing with lower cost, which will positively affect firm economic performance. On the other hand, the costs caused by the adoption of end-of-pipe technologies would not be effectively converted into market advantages. This, in turn, will make price-sensitive consumers less willing to buy and thus reduce economic performance. To test the theoretical mechanism that end-of-pipe technologies affect a firm’s economic performance through the role of green finance and green consumption, we hypothesize that a firm’s financial constraints and market advantages mediate the influence of environmental technologies on a firm’s economic outcomes. To summarize hypotheses H4a and H4b:

**Hypothesis** **4a** **(H4a):***Financial constraints mediate the relationship between end-of-pipe technologies and economic performance*.

**Hypothesis** **4b** **(H4b):***Market advantages mediate the relationship between end-of-pipe technologies and economic performance*.

Compared with end-of-pipe technologies, the adoption of clean technologies provides additional benefits from investors and consumers because clean technologies are more uncertain but more effective in the long run, and more in line with the preference of green investors. Meanwhile, more innovative clean technologies are more conducive to obtaining a first-mover advantage for green products. In light of increasing environmental awareness, the enterprises can a achieve comparison advantage in the financial market and product market by introducing clean technologies, which then improve their economic performance. Summarizing Hypotheses H4c and H4d:

**Hypothesis** **4c** **(H4c):***Financial constraints mediate the relationship between clean technologies and economic performance*.

**Hypothesis** **4d** **(H4d):***Market advantages mediate the relationship between clean technologies and economic performance*.

The overall research model and hypotheses are summarized in Figure 1 and Table 1.

## 3. Research Design

### 3.1. Data Sources

Our initial sample consists of all firms in manufacturing industries listed on the A-share market of the SSE (Shanghai Stock Exchange) and the SZSE (Shenzhen Stock Exchange) from 2011 to 2018. To ensure the completeness and reliability of our data analysis, the following guidelines were used in our sample selection: (1) Excluding ST and ST*firms to ensure the stability and validity of sample; (2) excluding firms established within three years; and (3) excluding firms with incomplete or missing financial information. Finally, we obtained a sample of 13,275 firm-years from 2123 manufacturing firms. The financial data for the firms were primarily obtained from the China Stock Market & Accounting Research (CSMAR) database. The patent data were mainly acquired from the Patsnap patent network (https://analytics.zhihuiya.com, accessed date: 7 April 2020), it was possible to identify individual companies that have been granted or have applied for a patent.

### 3.2. Variables

#### 3.2.1. Dependent Variable: Economic Performance

Return on assets (ROA) is a standard accounting measure of economic performance commonly used in the literature, indicating the outcomes of specific past and present actions [19]. ROA is more stable than sales growth or return on sales in measuring economic performance because of both the managerial effect of short-term activities and uncertainty about the external environment in emerging markets. Thus, because of its stability and reliability, we used ROA to measure the economic performance of the firms [22].

#### 3.2.2. Independent Variable: Environmental Technologies

In this study, environmental technologies were divided into end-of-pipe technologies and clean technologies [8,28,53]. We employed the firm’s capital expenditure in environment protection alteration to measure the firm’s end-of-pipe technologies (PIPE). The capital expenditure was hand-collected from the “construction in progress” notes in the firm annual report, including environmental treatment, sewage treatment, environmental design, and energy conservation, and waste recycling [54]. We used the ratio of the total amount of end-of-pipe technologies to total assets to measure the intensity of end-of-pipe technologies.

Considering the availability of data, we used the green patents applicated by enterprises to measure the level of clean technologies (PATENT). However, it must be clarified that this may underestimate the level of clean technology of enterprises, since enterprise clean technology also includes technology introduction. Based on the literature, we identified clean technologies by using the IPC Green Inventory, which were developed by the IPC Committee of Experts [55,56]. Due to the distribution of green patents being skewed, we referred to Li et al.’s study and employed the logarithm of the number of green patent applications to measure the firm’s clean technologies [57].

#### 3.2.3. Mediators

##### Financial Constraints

Based on previous literature, we employed the SA index (SA) to measure a firm’s financial constraints [58]. The calculation formula is as follows: SA = −0.737 × Size+0.043 × Size^2^ − 0.04 × Age, where Size is the log of total assets and Age is the number of years the enterprise has been listed. Taking into account that the SA index is negative; the greater its absolute value, the more serious the financial constraints [59].

##### Market Advantages

Market share (SHARE) can reflect the competition situation of the existing enterprises in the industry (horizontal competitiveness from the perspective of the same industry). We employed the firm’s market share to measure market advantages. The larger the market share is, the more favored the enterprise’s products are in the market, which also means that the enterprise’s market advantages are stronger [60].

#### 3.2.4. Controls

In order to accurately reflect the influence of the environmental technologies on economic performance, this paper also controls the firm size (SIZE), firm age (AGE), cash flow (CASH), the ratio of intangible assets (IA), organizational slack (SLACK), green subsidies (SUBSIDY), CEO duality (DUALITY), board independence (BI), and the largest shareholder’s shareholding (LS) (see details of the variable definition in Appendix A
Table A1).

### 3.3. Empirical Models

To assess whether the environmental technologies improve economic performance through the channel of mitigating financial constraints and improving market share, we conducted empirical research using a mediation effect test method [61]. First, to validate the relationship between the environmental technologies and the firm’s economic performance, based on H1a–H1b, Equation (1) is constructed.

Then, we established that environmental technologies are associated with the mediator (financial constraints and market share), based on H2a–H2b and H3a–H3b, Equations (2) and (3) are constructed.

Finally, to show what mediates the relationship between environmental technologies and dependent variables, we repeat the analysis for Equation (1) by adding the mediating as the additional independent variable, based on H4a–H4d, the Equation (4) is constructed. All equations are shown below: (1)$${ROA}_{it}=\alpha_{0}+\beta_{1}PIPE_{it}+\beta_{2}PATENT_{it}+\beta_{k}Controls_{it}+\varepsilon_{it}$$
(2)$$SA_{it}=\alpha_{0}+\beta_{1}PIPE_{it}+\beta_{2}PATENT_{it}+\beta_{k}Controls_{it}+\varepsilon_{it}$$
(3)$$SHARE_{it}=\alpha_{0}+\beta_{1}PIPE_{it}+\beta_{2}PATENT_{it}+\beta_{k}Controls_{it}+\varepsilon_{it}$$
(4)$$ROA_{it}=\alpha_{0}+\beta_{1}PIPE_{it}+\beta_{2}PATENT_{it}+\beta_{3}SA_{it}+\beta_{4}SHARE_{it}+\beta_{k}Controls_{it}+\varepsilon_{it}$$
where *i* denotes the firm, *t* indicates the year, *β* and *γ* represent estimated parameters, and *ε_it_* represents stochasticity across the firm.

## 4. Empirical Results

### 4.1. Descriptive Statistical Analysis

The summary statistics of the variables are shown in Table 2. The average value of ROA is 0.044, and the standard deviation is 0.076, indicating a difference in the economic performance between different companies. The median of PATENT is 0, while the median of PIPE is 0, indicating that over half of the manufacturing enterprises have not invested in environmental technologies. We also conducted an analysis to examine environmental technologies adoption by firms under different subgroups. For the group with ROA above the mean, 24 percent of the companies used clean technology, and 14 percent used end-of-pipe technology. There was no difference in the use of cleaning technologies compared to the group below the mean, but the use of end-of-pipe technologies was lower. Further analysis shows that the number of green patents is higher in the above-average group than in the low-average group. This analysis verifies that manufacturing companies that tend to adopt green technologies are more likely to achieve higher economic performance. The correlation coefficients of all variables are less than 0.5, indicating that there is no serious multi-collinearity among variables (Appendix A
Table A2 provides the full correlation matrix).

### 4.2. Regression Analysis

This paper uses the fixed effects model to verify the impact of environmental technologies on the economic performance of manufacturing firms, as well as the mediating role of financial constraints and market share. Table 3 provides the results for hypotheses (H1) to (H4). Model 1 examines the direct effects of end-of-pipe technologies and clean technologies on economic performance. The coefficient of PIPE is positive and significant (Model 1, β1 = 0.1592, *p* = 0.028), supporting H1a. Model 1 also shows that the coefficient for PATENT is significant and positive (Model 1, β2 = 0.0028, *p* = 0.032). This provides support for H1b. Ceteris paribus, the results indicate that if a manufacturing firm’s end-of-pipe technologies investment increases by 1% or the number of green patent applications increase by 1, the company will see a 0.159 and 0.0028% increase in ROA respectively. Therefore, it can be concluded that no matter whether manufacturing firms use more clean technologies or end-of-pipe technologies, they can obtain better economic performance.

Model 3 tests the direct effects of end-of-pipe technologies and clean technologies on financial constraints. The results indicate that firms with clean technologies which attracting green investment tend to have lower financial constraints (Model 2, β_2_ = −0.0074, *p* = 0.000). That is, ceteris paribus, if companies apply for an additional green patent, enterprises with average financial constraints level will see a 0.0074% decrease in financial constraints. This provides support for H2b. However, the effect of end-of-pipe technologies on financial constraints is not significant (Model 2, β_1_ = 0.0321, *p* = 0.482). H2a is not supported. In addition, Model 4 examines the direct effects of end-of-pipe technologies and clean technologies on market advantages. The result shows that manufacturing companies using end-of-pipe technology have a negative and significant association with market advantages (Model 3, β_1_ = −0.0369, *p* = 0.021). Ceteris paribus, a 1% increase in end-of-pipe technologies investment reduces ROA by 0.0369, supporting H3a. Meanwhile, there is no significant positive correlation between clean technologies and market advantage. H3b is not supported.

Models 4, 5, and 6 test the mediation effects. In Models 4 and 5, the coefficient of SA is negative and significant (Model 4, β_3_ = −0.0308, *p* = 0.041); the coefficient of SHARE is positive and significant (Model 5, β_4_ = −0.1920, *p* = 0.000), indicating that lower financial constraints and more market advantages enhance firms’ economic performance. When controlling for financial constraints and market advantages, the effect of end-of-pipe technologies on innovation performance is still significant and the coefficient of end-of-pipe technologies increases from 0.1592 to 0.1669 (Model 6, β_1_ = 0.1669, *p* = 0.022). Associated with the significant negative impact of end-of-pipe technologies on market advantages, these results indicate that market advantages mediate the relationship between end-of-pipe technologies and economic performance. Significantly, the mediation of market share is a masking effect, that is, the negative effect of end-of-pipe technologies on the market share will weaken the positive effect of end-of-pipe technologies on economic performance. H4b is supported, while H4a is not.

After controlling for mediators, the coefficient of PATENT is also significant and decreases from 0.0028 to 0.0026 (Model 6, β_2_ = 0.0026, *p* = 0.053), which indicated that financial constraints have a mediation effect on the relationship between clean technologies and a firm’s economic performance. In other words, clean technologies not only directly influence economic performance but also indirectly affects economic performance through mitigating financial constraints. H4c is supported, while H4d is not.

### 4.3. Robustness Testing

#### 4.3.1. Heckman Two-Stage Procedure

In the study above, this research preliminarily demonstrates the positive relationship between environmental technologies and economic performance. However, the relationship may be affected by unobservable variables, resulting in wrong results. To ensure the robustness of empirical results, this paper uses a two-stage processing effect model to analyze the impact of environmental technologies on a firm’s economic performance. The first-step regression was conducted to obtain the inverse Mill ratio (IMR). In the second-step regression, the IMR was introduced into all models. Table 4 shows the regression estimates of the above two steps. As shown in Panel A of Table 4, we conducted the probit model for the first-step regression, with the dependent variable was the environmental technologies adoption dummy (ETA), which equals 1 if the firm adopts environmental technologies, and 0 otherwise. Carbon markets are a globally accepted tool to encourage the adoption of environmental technologies [62]. China’s pilot carbon emissions trading programs began operating in the second half of 2013 in seven provinces and cities [63]. The new variable introduced in the first-step regression is the dummy variable of carbon emission trading pilots (CEP).

Through the above two-stage regression test, it can be found that in the first-stage regression results, there is a significant positive correlation between CEP and environmental technologies adoption. The result from Panel A indicates that manufacturing companies participating in Carbon Emission Trading are 26.71% more likely to adopt environmental technologies compared to non-pilot manufacturing companies. In the second stage regression, the inverse Mills ratio is significant in each model, which suggests that our models have a problem with sample selection bias. The two-stage regression results after controlling for endogenous selection bias show that there is a significant positive correlation between different types of environmental technologies and economic performance. These results are consistent with the above analysis conclusions.

#### 4.3.2. Accuracy of Independent Variable Measurement

The difference between clean technologies and end-of-pipe technologies is that the former affects the production process itself. Therefore, we can analyze the impact of different types of environmental technologies measurement indicators on enterprise productivity to demonstrate the accuracy of independent variable measurement. In this paper, we use a semi-parametric estimation approach (LP for short) to calculate the total factor productivity (TFP). We provide the results of the effect of end-of-pipe technologies and clean technologies on TFP in Table 4. The result shows that the effect of clean technologies on TFP is significant and positive (Model 1, β = 0.0166, *p* = 0.02). Ceteris paribus, if companies apply for an additional green patent, enterprises will see a 0.0167% increase in TFP. However, the coefficient of end-of-pipe technologies is not significant, indicating the end-of-pipe technologies has a limited impact on the firms’ productivity. Taken together, our use of environmental alteration investments and green patents to measure different types of environmental technologies is accurate, and only clean technologies play an important role in a firm’s productivity.

#### 4.3.3. One-Year Lagged Effect

As environmental technologies take a longer period of time to influence firm economic performance, we ran a robustness analysis by lagging environmental technologies by tone-year periods in Table 5. The result for clean technologies indicates that financial constraints and market advantages mediate the positive influence of PTENT_t−1_ on a firm’s economic performance, and the effects last for at least 1 year. On the other hand, the results for PIPE_t−1_ differ in that it is not significantly associated with economic performance. These results show that clean technologies have a more permanent impact on economic performance than end-of-pipe technologies.

## 5. Conclusions

This study explored the relationship between environmental technologies and economic performance using a sample of listed manufacturing companies. As environmental technologies are divided into clean technologies and clean technologies, both types of environmental technologies can improve firms’ economic performance. Concurrently, this study proved that the more focus a manufacturing firm on environmental technologies inputs, the more it is equipped to meet the green finance and green consumption demands; consequently, its economic performance is improved through mitigating financial constraints and improving market advantage. Therefore, we draw the following conclusions.

First, the adoption of environmental technologies, whether clean technologies or end-of-pipe technologies, could improve the economic performance of manufacturing firms. On one hand, environmental technologies could directly reduce pollution control costs and increase revenue. On the other hand, environmental technologies could indirectly affect economic performance through external benefits and resources from stakeholders. Second, clean technologies not only directly affect economic performance but also indirectly affects economic performance through mitigating financial constraints; end-of-pipe technologies do not significantly affect financial constraints. The possible explanation for this outcome is that end-of-pipe technologies need to pay high transformation costs in the short term, which cannot be covered by the support of green finance. Finally, the end-of-pipe technologies not only directly influence economic performance but also indirectly affect economic performance through weakened market advantages; clean technologies do not significantly affect market advantages. The reason may be the complexity in the process of environmental technologies compared to traditional innovation, which leads to the difficulty to forming market advantages quickly. In addition, the awareness of green consumption is still inadequate.

The aforementioned findings also significantly contribute to industrial implications and social practice. First, this study provided new ideas for green transformation of manufacturing industries. Most Chinese firms have a low environmental innovation capacity [7]. This phenomenon hinders the development of companies, especially for manufacturing enterprises. This paper found that the adoption of environmental technologies, whether end-of-pipe technologies or clean technologies, could improve a manufacturing firm’s economic performance. In addition, compared with end-of-pipe technologies, clean technologies will be more effective in the long run. Therefore, the implementation of green innovation and environmental disclosure should be enhanced to help firms obtain more financing resources and maximize benefits and ensure the rapid development of enterprises and healthy economic practices.

Second, this research provides important help for the government to more effectively formulate and implement green financial system. Green finance is an important pushing way of green transformation and development, it encourages enterprises to conduct environmental technologies and promotes the clean production of industries. Our study shows that the Chinese green financial system provides effective financial support for manufacturing enterprises adopting clean technologies, while it does not for those adopting end-of-pipe technologies. This requires the government to formulate more detailed financial policies and improve differentiated financial policies for enterprises adopting different types of environmental technologies. By promoting these practices, the governments could also alleviate the financing difficulties of manufacturing enterprises and then proceeding to promote regional green development economic growth.

Finally, our result stresses the importance of popularizing the concept of green consumption. On one hand, enterprises require an in-depth analysis of the market green demands and turn green R&D investment into a measurable business return. Meanwhile, the enterprises need to establish the overall green image of the enterprise through building green product promotion and sales channels. On the other hand, the government is the most important promoter of green consumption; it shoulders the responsibility of promoting and disseminating knowledge related to green environmental protection. By strengthening market supervision and fostering the market participants’ attention for green development, we could create a situation where the enterprise’s green product innovation ability and market shares are closely related and mutually reinforcing.

As with any empirical study, our study is not exempt from limitations. A primary limitation relates to the use of green patents to measure clean technologies. It should be recognized that we focused exclusively on green patents, which led to an underestimation of the impact of clean technologies. Future studies might focus on introducing clean technologies, which then can encompass overall R&D funding of clean technologies. Furthermore, future environmental technologies studies might compare the impact of different environmental technologies types on enterprise performance through the expenditure of capitalization. A second limitation is our use of a carbon trading pilot policy as an antecedent variable influencing the adoption of environmental technologies. We encourage future studies to collect different antecedent variables for specific types of environmental technologies to more effectively control model endogeneity. Another limitation is our focus on green financing and green consumption. An interesting direction for future studies would be to investigate the link between environmental technologies and benefits from other stakeholders. Finally, a potential limitation is the lack of examination of industry heterogeneity. We certainly encourage scholars to conduct comparative studies on different industries, which may provide more targeted guidance for enterprises to strengthen their environmental technologies and economic performance.

## Figures and Tables

**Figure 1 ijerph-18-04012-f001:**
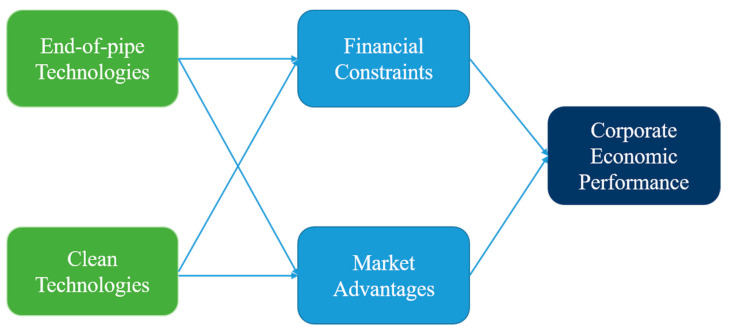
Research model.

**Table 1 ijerph-18-04012-t001:** Hypotheses summary.

Hypothesis	Mediation Notation
H1a	X → Y for end-of-pipe technologies
H1b	X → Y for clean technologies
H2a	X → M1 for end-of-pipe technologies
H2b	X → M1 for clean technologies
H3a	X → M2 for end-of-pipe technologies
H3b	X → M2 for clean technologies
H4a	X → M1 → Y for end-of-pipe technologies
H4b	X → M2 → Y for end-of-pipe technologies
H4c	X → M1 → Y for clean technologies
H4d	X → M2 → Y for clean technologies

Note: X, M1, M2, and Y represent the following notations: X—independent variable; M1—mediating variable financial constraints; M2—mediating variable market advantages; Y—dependent variable.

**Table 2 ijerph-18-04012-t002:** Descriptive statistics.

	(1)	(2)	(3)	(4)	(5)	(6)
Variables	Observation	Mean	SD	Min	Median	Max
ROA	13,275	0.0442	0.0764	−2.008	0.042	2.163
PIPE	13,275	0.00128	0.00877	0	0	0.357
PATENT	13,275	0.370	0.801	0	0	6.430
SA	13,275	3.456	0.288	2.593	3.413	4.294
SHARE	13,275	1.780	4.881	0.000587	0.378	79.47
SIZE	13,275	21.89	1.153	18.29	21.747	27.39
AGE	13,275	15.76	5.488	4	15	51
CASH	13,275	0.0455	0.0708	−1.080	0.044	0.661
IA	13,275	0.0464	0.0408	0	0.038	0.677
SLACK	13,275	0.649	0.564	0.00488	0.555	21.36
SUBSIDY	13,275	4.631	6.419	0	0	22.00
DUALITY	13,275	0.300	0.458	0	0	1
BI	13,275	0.373	0.0547	0.143	0.333	0.800
LS	13,275	34.72	14.36	3.003	33.145	89.99

ROA: Return on assets; PIPE: firm’s end-of-pipe technologies; PATENT: green patents applicated by enterprises to measure the level of clean technologies; SA: SA index; SHARE: Market share; SIZE: firm size; AGE: firm age; CASH: cash flow; IA: the ratio of intangible assets; SLACK: organizational slack; SUBSIDY: green subsidies; DUALITY: CEO duality; BI: board independence; LS: largest shareholder’s shareholding.

**Table 3 ijerph-18-04012-t003:** Baseline results of overall samples.

	(1)	(2)	(3)	(4)	(5)	(6)
	ROA	SA	SHARE	ROA	ROA	ROA
PIPE	0.1592 **	0.0321	−3.6866 **	0.1601 **	0.1662 **	0.1669 **
	(2.1912)	(0.7026)	(−2.3031)	(2.2051)	(2.2901)	(2.2990)
PATENT	0.0028 **	−0.0074 ***	0.0426	0.0026 **	0.0028 **	0.0026 *
	(2.1481)	(−8.8726)	(1.4634)	(1.9697)	(2.0878)	(1.9374)
SA				−0.0308 **		−0.0260 *
				(−2.0432)		(−1.7264)
SHARE					0.0019 ***	0.0019 ***
					(4.4690)	(4.3330)
SIZE	0.0167 ***	0.0434 ***	0.8410 ***	0.0180 ***	0.0150 ***	0.0162 ***
	(9.8196)	(40.6937)	(22.5028)	(9.8984)	(8.6797)	(8.7101)
AGE	−0.0063 ***	0.0383 ***	−0.2054 ***	−0.0051 ***	−0.0059 ***	−0.0049 ***
	(−15.5037)	(1.5e+02)	(−22.8394)	(−7.2811)	(−14.2203)	(−6.9852)
CASH	0.1184 ***	−0.0030	0.7527 ***	0.1183 ***	0.1169 ***	0.1169 ***
	(11.4797)	(−0.4580)	(3.3119)	(11.4724)	(11.3436)	(11.3410)
IA	−0.0754 ***	−0.0447 ***	0.4630	−0.0768 ***	−0.0763 ***	−0.0775 ***
	(−3.4140)	(−3.2174)	(0.9507)	(−3.4751)	(−3.4570)	(−3.5074)
SLACK	−0.0514***	0.0014	0.0227	−0.0514 ***	−0.0515 ***	−0.0515 ***
	(−33.8679)	(1.5167)	(0.6789)	(−33.8398)	(−33.9248)	(−33.8984)
SUBSIDY	−0.0001	0.0002 ***	−0.0029	−0.0001	−0.0001	−0.0001
	(−0.5552)	(2.8747)	(−1.0799)	(−0.4994)	(−0.5099)	(−0.4639)
DUALITY	0.0069 ***	−0.0035 **	−0.0149	0.0068 ***	0.0070 ***	0.0069 ***
	(3.2103)	(−2.5376)	(−0.3118)	(3.1607)	(3.2262)	(3.1835)
BI	−0.0195	−0.0013	0.8077 **	−0.0196	−0.0211	−0.0211
	(−1.1139)	(−0.1168)	(2.0903)	(−1.1163)	(−1.2032)	(−1.2027)
LS	0.0006 ***	−0.0005 ***	0.0009	0.0006 ***	0.0006 ***	0.0006 ***
	(5.7657)	(−6.7588)	(0.3537)	(5.6242)	(5.7556)	(5.6342)
Fe	Y	Y	Y	Y	Y	Y
Year	Y	Y	Y	Y	Y	Y
Constant	−0.2081 ***	−1.9206 ***	−13.6908 ***	−0.1490 ***	−0.1818 ***	−0.1325 ***
	(−5.9191)	(−86.8050)	(−17.6693)	(−3.2737)	(−5.1047)	(−2.9037)
*N*	13275	13275	13275	13275	13275	13275
R^2^	0.1359	0.8900	0.0633	0.1362	0.1374	0.1377

Note: * *p* < 0.1, ** *p* < 0.05, *** *p* < 0.01.

**Table 4 ijerph-18-04012-t004:** Results with Heckman two-stage procedure.

Panel A: The First−Step Regression—Model Employed to Estimate Inverse Mills									
Variable	CEP	AGE	CASH	IA	SUBSIDY	Year	Constant	N	R^2^
ETA	0.2671 ***	−0.0088 ***	0.0092	−1.0467 ***	0.0259 ***	Y	−0.5036 ***	13,275	0.0196
	−5.81	(−3.94)	−0.06	(−3.64)	−14.71		(−9.17)		
Panel B: The Second−Step Regression—After Introducing Inverse Mills									
		−1	−2	−3	−4	−5		−6	
		ROA	SA	SHARE	ROA	ROA		ROA	
PIPE		0.1622 **	0.0187	−3.7564 **	0.1627 **	0.1695 **		0.1697 **	
		−2.2329	−0.4136	(−2.3469)	−2.2398	−2.3349		−2.338	
PATENT		0.0028 **	−0.0072 ***	0.0435	0.0026 **	0.0027 **		0.0026 *	
		−2.1209	(−8.7814)	−1.4924	−1.9717	−2.0585		−1.9391	
SA					−0.0261*			−0.0209	
					(−1.7124)			(−1.3712)	
SHARE						0.0019 ***		0.0019 ***	
						−4.5233		−4.4051	
IMR		−0.0471 **	0.2090***	1.0869 **	−0.0416 **	−0.0492 **		−0.0448 **	
		(−2.2922)	−16.3577	−2.4011	(−2.0030)	(−2.3965)		(−2.1546)	
SIZE		0.0166 ***	0.0438 ***	0.8431 ***	0.0177 ***	0.0149 ***		0.0159 ***	
		−9.7653	−41.5486	−22.5576	−9.7144	−8.6122		−8.5019	
AGE		−0.0074 ***	0.0431 ***	−0.1807 ***	−0.0063 ***	−0.0070 ***		−0.0061 ***	
		(−11.9351)	−111.7797	(−13.2339)	(−6.9480)	(−11.2897)		(−6.8165)	
CASH		0.1181 ***	−0.0019	0.7585 ***	0.1181 ***	0.1167 ***		0.1167 ***	
		−11.457	(−0.2901)	−3.3378	−11.4532	−11.3182		−11.3182	
IA		−0.0393	−0.2054 ***	−0.3722	−0.0446	−0.0385		−0.0429	
		(−1.4459)	(−12.1583)	(−0.6221)	(−1.6325)	(−1.4204)		(−1.5690)	
SLACK		−0.0515 ***	0.0018 *	0.0247	−0.0515 ***	−0.0516 ***		−0.0515 ***	
		(−33.9196)	−1.9264	−0.7364	(−33.8856)	(−33.9799)		(−33.9494)	
SUBSIDY		−0.0009 **	0.0041 ***	0.0172 **	−0.0008 **	−0.0010 **		−0.0009 **	
		(−2.3529)	−16.4724	−1.9627	(−2.0608)	(−2.4386)		(−2.1949)	
DUALITY		0.0069 ***	−0.0031 **	−0.013	0.0068 ***	0.0069 ***		0.0068 ***	
		−3.1735	(−2.3036)	(−0.2731)	−3.1356	−3.188		−3.157	
BI		−0.0189	−0.0042	0.7925 **	−0.019	−0.0204		−0.0205	
		(−1.0763)	(−0.3869)	−2.051	(−1.0827)	(−1.1650)		(−1.1680)	
LS		0.0006 ***	−0.0004 ***	0.0011	0.0006 ***	0.0006 ***		0.0006 ***	
		−5.671	(−6.1825)	−0.4491	−5.5616	−5.6567		−5.5672	
Fe		Y	Y	Y	Y	Y		Y	
Year		Y	Y	Y	Y	Y		Y	
Constant		−0.1366 ***	1.6034 ***	−15.3405 ***	−0.0948 *	−0.1068 **		−0.074	
		(−2.9082)	−54.8612	(−14.8152)	(−1.7905)	(−2.2535)		(−1.3922)	
N		13275	13275	13275	13275	13275		13275	
R^2^		0.1363	0.8926	0.0638	0.1365	0.1379		0.138	

Note: * *p* < 0.1, ** *p* < 0.05, *** *p* < 0.01.

**Table 5 ijerph-18-04012-t005:** Results of other robustness tests.

	(1)	(2)	(3)	(4)	(5)
	TFP	F.ROA	F.SA	F.SHARE	F.ROA
PIPE	−0.1460	0.0130	0.0854 *	−0.7503	0.0088
	(−0.4993)	(0.1601)	(1.9026)	(−0.5352)	(0.1079)
PATENT	0.0167 ***	0.0028*	−0.0073 ***	0.0477 *	0.0033 **
	(3.0321)	(1.7845)	(−8.3146)	(1.7341)	(2.0446)
F.SA					0.0790 ***
					(4.1200)
F.SHARE					0.0033 ***
					(5.3771)
SIZE	0.5476 ***	−0.0203 ***	0.0185 ***	0.5985 ***	−0.0237 ***
	(72.8001)	(−9.5871)	(15.8436)	(16.4150)	(−10.9036)
AGE	−0.0094 ***	−0.0009*	0.0400 ***	−0.1489 ***	−0.0036 ***
	(−5.3053)	(−1.7154)	(137.9196)	(−16.4470)	(−3.8414)
CASH	0.6867 ***	0.1046 ***	−0.0017	0.1769	0.1041 ***
	(15.4177)	(8.5233)	(−0.2546)	(0.8377)	(8.5061)
IA	−0.7567 ***	−0.1143 ***	0.0006	−0.2563	−0.1135 ***
	(−7.9813)	(−4.3214)	(0.0429)	(−0.5632)	(−4.3012)
SLACK	−0.1000 ***	0.0302 ***	−0.0007	0.0482	0.0301 ***
	(−12.1477)	(17.0459)	(−0.7000)	(1.5788)	(17.0233)
SUBSIDY	0.0007	0.0001	0.0001	0.0007	0.0000
	(1.4827)	(0.3515)	(1.1737)	(0.2917)	(0.2845)
DUALITY	0.0158 *	0.0058 **	−0.0039 ***	−0.0266	0.0062 **
	(1.7157)	(2.1993)	(−2.6591)	(−0.5822)	(2.3526)
BI	−0.0503	−0.0005	0.0032	0.4656	−0.0023
	(−0.6847)	(−0.0234)	(0.2705)	(1.2579)	(−0.1069)
LS	0.0008 *	0.0006 ***	−0.0003 ***	0.0035	0.0006 ***
	(1.7083)	(4.1984)	(−3.9200)	(1.4520)	(4.2924)
Fe	Y	Y	Y	Y	Y
Year	Y	Y	Y	Y	Y
Constant	−3.8264 ***	0.4539 ***	2.4917 ***	−9.3514 ***	0.2880 ***
	(−24.3270)	(10.3974)	(103.5258)	(−12.4451)	(4.4558)
N	11694	11002	11002	11002	11002
R^2^	0.4982	0.0646	0.8744	0.0450	0.0691

Note: * *p* < 0.1, ** *p* < 0.05, *** *p* < 0.01.

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
