# Peer review of "Are Clean Technologies More Effective Than End-of-Pipe Technologies? Evidence from Chinese Manufacturing"

_ijerph, 2021, doi:10.3390/ijerph18084012_

Round 1

Reviewer 1 Report

1. The paper “Are clean technologies more effective than end-of-pipe technologies? is interesting. 

This research show that end-of-pipe technologies and clean technologies have a positive effect on firms' economic performance. Moreover, we find that clean technologies not only directly affect economic performance but also indirectly affects economic performance through mitigating financial constraints. With the negative influence of end-of-pipe technologies on market advantages, the positive economic effect caused by end-of-pipe technologies is weakened. This research provides useful insights into the selection of environmental technologies for manufacturing firms and the establishment of new policies to promote green finance and green consumption. 

  1. The paper should cite some literatures in the introduction, for example the following literatures.

Assessing the policy impacts on non-ferrous metals industry's CO2 reduction: Evidence from China, Journal of Cleaner Production, Volume 192, 10 August 2018, 252-261

The volatility spillover effect of the European Union (EU) carbon financial market, Journal of Cleaner Production,Volume 282, 1 February 2021, 124394.

Structural decline in China's CO2 emissions through transitions in industry and energy systems. Nature Geoscience. 2018,11(8): 551

The response of the Beijing carbon emissions allowance price (BJC) to macroeconomic and energy price indices,          Energy Policy, 2017,106: 111-121.

  1. The extensive editing of English language and style is required.

4. I think it is suitable for publication. after revision.

Author Response

Dear Reviewer:

We appreciate you very much for the constructive comments and suggestions on our manuscript. The following is a detailed description of how we address your concerns.

1.1. Summary

RC: The paper “Are clean technologies more effective than end-of-pipe technologies? ” is interesting. This research show that end-of-pipe technologies and clean technologies have a positive effect on firms' economic performance. Moreover, we find that clean technologies not only directly affect economic performance but also indirectly affects economic performance through mitigating financial constraints. With the negative influence of end-of-pipe technologies on market advantages, the positive economic effect caused by end-of-pipe technologies is weakened. This research provides useful insights into the selection of environmental technologies for manufacturing firms and the establishment of new policies to promote green finance and green consumption.

  • Comments #1

RC: The paper should cite some literatures in the introduction, for example the following literatures.

Assessing the policy impacts on non-ferrous metals industry's CO2 reduction: Evidence from China, Journal of Cleaner Production, Volume 192, 10 August 2018, 252-261

The volatility spillover effect of the European Union (EU) carbon financial market Journal of Cleaner ProductionVolume 282, 1 February 2021, 124394.

Structural decline in China's CO2 emissions through transitions in industry and energy systems. Nature Geoscience. 2018,11(8): 551

The response of the Beijing carbon emissions allowance price (BJC) to macroeconomic and energy price indices, Energy Policy, 2017,106: 111-121.

AR: We thank the reviewer for recommending these useful references. These references greatly enriched our understanding. All the references are cited in our revised manuscript (Page 19, REFERENCES).

  • Comments #2

RC: The extensive editing of English language and style is required.

AR: We thank the reviewer for this useful suggestion, and we apologize for the writing problems in the article. For this reason, we have rewritten parts of the introduction, hypothesis and analysis of results (Page 2, section 1, lines 81-91; Page 5, section 2.4, lines 228-258; Page 1, section 4.2, lines 444-473).

  • Comments #3

RC: I think it is suitable for publication. after revision.

AR: We appreciate your careful reading and constructive suggestions. These comments are of great help in improving this paper. We have tried our best to revise the manuscript according to your comments. We hope that you are satisfied with the revised manuscript.

Reviewer 2 Report

  1. Several writing mistakes throughout the text. For example, writing “china” instead of “China.” Some corrections are suggested in the attached file.
  2. When the authors wrote “oxynitride” they might have meant “nitrogen oxides.”
  3. The authors brought up the “stakeholder theory... regarding the investors and customers.” On p. 2. It might be advisable to put down a reference at least about it.
  4. It seems that there are too many hypotheses. Fig. 1 might look simple but at a glance, you cannot tell what each hypothesis means.
  5. On p. 7, the authors wrote “We employ the logarithm of the number of green patent applications to measure the firm’s clean technologies.” Why logarithms?
  6. Apparently, a reference is missing in line 293.
  7. The technical procedure looks sound. However, the explanation is rather poor. The tables should be as self-explanatory as possible.
  8. The technical procedure might be at the end as an appendix to make the reading more fluent and highlight the connection of the results with the hypotheses.
  9. The conclusions should also deal with how the hypotheses relate to what is stated.
  10. The study is ambitious and seems valuable. It also has good theoretical grounding. Nevertheless, the results are not very clear and the paper fails to establish a clear relation with the theory and hypotheses.

Author Response

Dear Reviewer:

We appreciate you very much for the constructive comments and suggestions on our manuscript entitled “Are clean technologies more effective than end-of-pipe technologies? Evidence from Chinese manufacturing"(ID:ijerph-1113102).

These comments are all valuable and helpful for improving our article. All the authors have seriously discussed all these comments. We hope that you are satisfied with the revision. However, we would be more than willing to make any further changes that you deem necessary. The following is a detailed description of how we address  your concerns. If it is not clear you can view the attachment.

  • Comments #1

RC: Several writing mistakes throughout the text. For example, writing “china” instead of “China.” Some corrections are suggested in the attached file.

AR: We apologize for this mistake. We have revised the mistake in our updated draft. (Page1, section 1, line 31)

1. Introduction

…the largest energy consumption industry in china China’s economic development…

  • Comments #2

RC: When the authors wrote “oxynitride” they might have meant “nitrogen oxides.”

AR: We apologize for the imprecise statement and thank the reviewer for pointing out this. We have changed “oxynitride” to “nitrogen oxides” in the revised draft (Page 1, section 1, line 37).

1. Introduction

…the more than 38.68% of oxynitride nitrogen oxides emissions comes from manufacturing…

  • Comments #3

RC: The authors brought up the “stakeholder theory... regarding the investors and customers.” On p. 2. It might be advisable to put down a reference at least about it.

AR: We thank the reviewer for this helpful advice. In the revised manuscript, we further added to the relevant content and literature on stakeholder theory in the introduction (Page 1, section 1, lines 81-84). The following are the contents of the additions based on this comment.

1. Introduction

In recent years, more and more scholars have examined the benefits of environmental technologies from external stakeholders, such as investors, based on stakeholder theory [13,14]. Along with green finance and green consumption spring up, stakeholders are playing an increasing role in the profits of manufacturing enterprise.  

  • Comments #4

RC: It seems that there are too many hypotheses. Fig. 1 might look simple but at a glance, you cannot tell what each hypothesis means.

AR: We apologize for the confusion in this figure. In the updated manuscript, we have simplified the model diagram and added Table 1 to explain each assumption (Page 7, section 2.4, lines 286-292). The following is the revised Figure 1 and added Table 1 according to this comment.

Figure 1 Research model

Table 1. Hypotheses summary

Hypothesis

Mediation notation

H1a

X → Y for end-of-pipe technologies

H1b

X → Y for clean technologies

H2a

X → M1 for end-of-pipe technologies

H2b

X → M1 for clean technologies

H3a

X → M2 for end-of-pipe technologies

H3b

X → M2 for clean technologies

H4a

X → M1→Y for end-of-pipe technologies

H4b

X →M2→ Y for end-of-pipe technologies

H4c

X → M1→Y for clean technologies

H4d

X →M2→ Y for clean technologies

Note: X, M1, M2 and Y represent the following notations: X—independent variable; M1—mediating variable financial constraints; M2—mediating variable market advantages; Y—dependent variable.

  • Comments #5

RC: On p. 7, the authors wrote “We employ the logarithm of the number of green patent applications to measure the firm’s clean technologies.” Why logarithms?

AR: We apologize for the lack of explanations on the logarithms and thank the reviewer for pointing out this issue. The distribution of green patents is skewed, with most of the data concentrated in the left-skewed part. For this reason, the logarithm of the patent data can transform it into a normal distribution. In the updated manuscript, we have added more explanation about the logarithms and related citations. The following is the revised content based on this comment (Page 8, section 3.2.2, lines 331–333).

3.2.2. Independent Variable: Environmental Technologies

Due to the distribution of green patents was skewed, we refer to Li et al.'s study and employ the logarithm of the number of green patent applications to measure the firm’s clean technologies [57]  

  • Comments #6

RC: Apparently, a reference is missing in line 293.

AR: We apologize for the errors in citation in our original draft and thank the reviewer for his or her careful reading (Page 15, section 3.3, paragraph 1, line 294). We revised the manuscript and added the references accordingly.

3.3. Empirical Models

…we conduct an empirical research using a mediation effect test method [61].  

  • Comments #7

RC: The technical procedure looks sound. However, the explanation is rather poor. The tables should be as self-explanatory as possible.

AR: We appreciate the reviewer for his or her careful reading and suggestions. We sincerely

apologize for not clearly articulating the results. In the revised manuscript, we have improved the presentation of the empirical results and added the corresponding explanations (Page 17, section 4.2, lines 444-473). The following is part of revised content based on this comment.

4.2. Regression Analysis

Model 3 tests the direct effects of end-of pipe technologies and clean technologies on financial constraints. The results indicate that firms with clean technologies which attracting green investment tend to have lower financial constraints (Model 2, β2 = -0.0074, p = 0.000). This provides support for H2b. However, the effect of end-of-pipe technologies on financial constraints is not significant (Model 2, β1=0.0321, p=0.482). H2a is not supported. In addition, model 4 examines the direct effects of end-of-pipe technologies and clean technologies on market advantages. The results show that manufacturing companies using end-of-pipe technology have a negative and significant association with market advantages (Model 3, β1 =-0.0369, p=0.021), supporting H3a. Meanwhile, there is no significant positive correlation between clean technologies and market advantage. H3b is not supported.  

  • Comments #8

RC: The technical procedure might be at the end as an appendix to make the reading more fluent and highlight the connection of the results with the hypotheses.

AR: We thank the reviewer for this very useful suggestion. Based on this comment, we have improved the original draft from two aspects. On the one hand, we have moved Table 1 (Variable definitions) and Table 3 (Pearson correlation matrix) from the original draft to the appendix section to enhance readability (Page 18, APPENDIX). On the other hand, we adjusted the order of hypothesis 4 so that the logical relationship between the analysis of results and each hypothesis corresponds more (Page 5, section 2.4, lines 232-257).

  • Comments #9

RC: The conclusions should also deal with how the hypotheses relate to what is stated.

AR: We appreciate the reviewer for raising this question. From our point of view, because of the large number of hypotheses in this paper, the second paragraph of the conclusion would be too lengthy if it were analyzed exactly in the order of the hypotheses. Therefore, we chose to elaborate the results of the hypotheses according to the type of environmental technologies.

  • Comments #10

RC: The study is ambitious and seems valuable. It also has good theoretical grounding. Nevertheless, the results are not very clear and the paper fails to establish a clear relation with the theory and hypotheses.

AR: We appreciate the reviewer’s careful reading and comment, and once again apologize for the problem in our original manuscript. In the revised manuscript, we have improved on the flawed logic between results and assumptions from two aspects. On the one hand, we strengthen the theoretical explanation of stakeholder theory (Page 2, section 1, lines 82-85; Page 4, section 2.2, lines 158-159; Page 4, section 2.3, lines 185-186). On the other hand, we improved the readability of results analysis by increasing notes and theoretical explanations (Page 12, section 4.2, lines 450-479).

Reviewer 3 Report

Based on an empirical test in some of Shangai and Shenzen companies (data from 2011 to 2018), this paper proposes a study of clean technologies and end-to-pipe technologies on company economic performance. the paper provides insight for selecting environmental technologies for manufacturing systems in order to promote green finance and green consumption.

This paper focuses attention on environmental technologies types and defines investment in environment protection alteration and green patent as indicators to reflect technological characteristics of end-of-pipe technologies and clean technologies.  

The authors confirms that environmental technologies may improve company economic performance. The paper indicates that more benefits could be obtained for a company by using green finance and green consumption through environmental technologies. New guidance and recommendations have been elaborated in this context.

Good paper ! 

But the following points need more attention! 

Lines 29 and 31: add spaces after China and problems. Do the same on lines 50, 51 etc.

Line 35 please define GDP: Gross domestic product?

The sentence from line 62 to 64 has to be improved: please change one “improve” term

The date 2010 written in line 73 is not the same as in the abstract.

The sentence from line 87 to 89 has to be improved (may be the term of performance has to be added).

Good introduction

Line 106 : may be it could nice to complete Yan et al. with the reference 21.

Line 113: change one reduced

Line 126 : space to be deleted

Line 146 may be “an” could be added before important.

Line 238: SSE and SZSE have to be explained.

A figure could be integrated in the research design for facilitating the understanding of the variables impacts and links, and then explaining and describing the model.

The part 4.1 could be improved by giving more explanation of the objectives of the analysis.

Line 321 : bad number of paragraph.

In part 4.3.2 the paragraph has to be detailed for facilitating the understanding of the analysis.

Part 4.3.3 has to be developed.

Good conclusion

Author Response

Dear Reviewer:

We appreciate you very much for the constructive comments and suggestions on our manuscript entitled “Are clean technologies more effective than end-of-pipe technologies? Evidence from Chinese manufacturing"(ID:ijerph-1113102).

These comments are all valuable and helpful for improving our article. All the authors have seriously discussed all these comments. We hope that you are satisfied with the revision. However, we would be more than willing to make any further changes that you deem necessary. The following is a detailed description of how we address your concerns. If it is not clear you can view the attachment.

1.1. Summary

RC: Based on an empirical test in some of Shangai and Shenzen companies (data from 2011 to 2018), this paper proposes a study of clean technologies and end-to-pipe technologies on company economic performance. the paper provides insight for selecting environmental technologies for manufacturing systems in order to promote green finance and green consumption.

This paper focuses attention on environmental technologies types and defines investment in environment protection alteration and green patent as indicators to reflect technological characteristics of end-of-pipe technologies and clean technologies. 

The authors confirms that environmental technologies may improve company economic performance. The paper indicates that more benefits could be obtained for a company by using green finance and green consumption through environmental technologies. New guidance and recommendations have been elaborated in this context.

Good paper !

  • Comments #1

RC: Lines 29 and 31: add spaces after China and problems. Do the same on lines 50, 51 etc.

AR: We apologize for the mistake in the citation format and appreciate the reviewer for pointing out this issue. We have revised the mistake in our updated draft (Page 1, section 1, lines 43, 56 and 58). In addition, we thoroughly checked the original manuscript and modified several similar errors.

  • Comments #2

RC: Line 35 please define GDP: Gross domestic product?

AR: We apologize for the confusion caused by this abbreviation. We have now added relevant explanations (Page 1, section 1, line 37).

1. Introduction

…The pure pursuit of GDP (Gross domestic product) at the expense of resource waste and environmental damage is bound to be unsustainable…

  • Comments #3

RC: The sentence from line 62 to 64 has to be improved: please change one “improve” term.

AR: We thank the reviewer for pointing out this mistake. We have revised that according to reviewer’s advice (Page 2, section 1, line 60).

1. Introduction

…environmental technologies can prompt the organization to reduce production cost and improve enhance processes and product to improve economic performance…

  • Comments #4

RC: The date 2010 written in line 73 is not the same as in the abstract.

AR: We apologize for this mistake. We have revised the mistake in our updated draft (Page 2, section 1, line 96).

1. Introduction

…a large firm-level dynamic panel data set from 20102011 to 2018 to investigate the effect…

  • Comments #5

RC: The sentence from line 87 to 89 has to be improved (may be the term of performance has to be added).

AR: We thank the reviewer for pointing out the errors and have revised the draft accordingly (Page 3, section 1, line 111).

1. Introduction

…environmental technologies may improve firms’ economic performance through alleviating financing constraints and market advantages…

  • Comments #6

RC: Line 106 : may be it could nice to complete Yan et al. with the reference 21.

AR: We appreciate the reviewer for his or her careful reading and suggestions. We have re-edited our presentation of reference 21 (Page 3, section 2.1, line 132).

2.1. Environmental Technologies and Economic Performance   

…Yan et al. found that both technology- and process-based environmental innovations positively influence airlines’ revenue…

  • Comments #7

RC: Line 113: change one reduced

AR: We appreciate the reviewer for this suggestion. We have changed “reduced” to “decrease” in the revised draft (Page 3, section 2.1, line 138).

2.1. Environmental Technologies and Economic Performance

…pollution control costs reduced decreased by reducing pollution emissions and waste recycling…

  • Comments #8

RC: Line 126 : space to be deleted

AR: We appreciate the reviewer’s careful reading, and have revised the mistake.

  • Comments #9

RC: Line 146 may be “an” could be added before important.

AR: We apologize for the errors in our original draft and thank the reviewer for his or her careful reading (Page 4, section 2.2, line 172).

2.2. Environmental Technologies and Financial constraints

…the environmental technologies are an important approach to reduce financial constraints…

  • Comments #10

RC: Line 238: SSE and SZSE have to be explained.

AR: We apologize for the lack of an explanation for SSE and SZSE. We revised the manuscript and added the explanations accordingly (Page 8, section 3.1, line 303).

3.1. Data Sources

…manufacturing industries listed on the A-share market of the SSE (Shanghai Stock Exchange) and the SZSE (Shenzhen Stock Exchange) …

  • Comments #11

RC: A figure could be integrated in the research design for facilitating the understanding of the variables impacts and links, and then explaining and describing the model.

AR: We apologize for the confusion in this figure. In the updated manuscript, we have simplified the model diagram and added Table 1 to explain each assumption (Page 7, section 2.4, lines 286-292). The following is the revised Figure 1 and added Table 1 according to this comment.

Figure 1 Research model

Table 1. Hypotheses summary

Hypothesis

Mediation notation

H1a

X → Y for end-of-pipe technologies

H1b

X → Y for clean technologies

H2a

X → M1 for end-of-pipe technologies

H2b

X → M1 for clean technologies

H3a

X → M2 for end-of-pipe technologies

H3b

X → M2 for clean technologies

H4a

X → M1→Y for end-of-pipe technologies

H4b

X →M2→ Y for end-of-pipe technologies

H4c

X → M1→Y for clean technologies

H4d

X →M2→ Y for clean technologies

Note: X, M1, M2 and Y represent the following notations: X—independent variable; M1—mediating variable financial constraints; M2—mediating variable market advantages; Y—dependent variable.

  • Comments #12

RC: The part 4.1 could be improved by giving more explanation of the objectives of the analysis.

AR: We apologize for the lack of explanations on statistical analysis and appreciate the reviewer for raising this suggestion. We entirely agree with the reviewer on his or her point of view. In the updated manuscript, we have added more explanation. The following are the contents of the additions based on this comment (Page 10, section 4.1, lines 386-394).

4.1 Descriptive Statistical Analysis

…We also conducted an analysis to examine environmental technologies adoption by firms under different subgroups. For the group with ROA above the mean, twenty-four percent of the companies used clean technology and fourteen percent used end-of-pipe technology. There was no difference in the use of cleaning technologies compared to the group below the mean, but the use of end-of-pipe technologies was lower. Further analysis shows that the number of green patents is higher in the above-average group than in the low-average group. This analysis verifies that manufacturing companies that tend to adopt green technologies are more likely to achieve higher economic performance.

  • Comments #13

RC: Line 321 : bad number of paragraph.

AR: Again, we apologize for the errors in our original draft and thank the reviewer for his or her careful reading (Page 11, section 4.2, line 402).

  • Comments #14

RC: In part 4.3.2 the paragraph has to be detailed for facilitating the understanding of the analysis.

AR: We apologize for the lack of explanations. In the revised manuscript, a more detailed explanation of the part 4.3.2 is given. The following are the contents of the additions based on this comment (Page 15, section 4.3.2, lines 521-528).

4.3.2 Accuracy of Independent Variable Measurement

…We provide the results of test for effect of end-of-pipe technologies and clean technologies on TFP in Table 4. The results show that the effect of clean technologies on TFP is significant and positive (Model 1, β=0.0166, p=0.02). However, the coefficient of end-of-pipe technologies is not significantly, indicating the end-of-pipe technologies has limited impact on the firms' productivity. Taken together, our use of environmental alteration investments and green patents to measure different types of environ-mental technologies is accurate, and only clean technologies play an important role in the firms' productivity…

  • Comments #15

RC: Part 4.3.3 has to be developed.

AR: We apologize for the confusion in this part. In the updated manuscript, we have added an explanation for the lagged effect. The following is the revised content according to this comment (Page 11, section 4.3.3, lines 530-538).

4.3.3 One-year lagged effect

As environmental technologies take a longer period of time to influence firm eco-nomic performance, we ran a robustness analysis by lagging environmental technologies by tone‐year periods in Table 5. The result for clean technologies indicates that financial constraints and market advantages mediate the positive influence of PTENTt−1 on firm's economic performance, and the effects last for at least 1 year. On the other hand, the results for PIPEt-1 differ in that it is not significantly associated with economic performance. These results show that clean technologies have a more permanent impact on economic performance than end-of-pipe technologies.

Round 2

Reviewer 2 Report

New comments:

  1. There was a marked improvement in the writing, which makes it easier to follow.
  2. The previous recommendations were attended, too.
  3. Unfortunately, the authors bring up “stakeholder theory” but has not articulated a basic explanation of it. First of all, they state that investors are considered as external stakeholders. It would be advisable to briefly explain this theory, or at least the meaning and relevance of “stakeholders,” according to it.  The references put down (13, 14) are not treatises on this theory.
  4. The change for Figure 1 makes it much more understandable. Table 1 for the hypotheses seems to contribute to keep them in mind and explain the variables, as well.
  5. “Models 1” (line 411) should read “Model 1.” This is written in blue, which are the corrections or additions. There are more mistakes, though.
  6. In Tables 3, 4 and 5, there should be an explanation of the parentheses and the relationship with the models. This in order to better the understanding of the information.

Author Response

Dear reviewer:

We thank you very much for giving us an opportunity to revise our manuscript. We hope that you are satisfied with the revision. However, we would be more than willing to make any further changes that you deem necessary. The following is a detailed description of how we address your concerns.

  • Comments #1

RC: There was a marked improvement in the writing, which makes it easier to follow.

AR: We appreciate your careful reading and constructive suggestions. The previous comments are of great help in improving this paper.

  • Comments #2

RC: The previous recommendations were attended, too.

  • Comments #3

RC: Unfortunately, the authors bring up “stakeholder theory” but has not articulated a basic explanation of it. First of all, they state that investors are considered as external stakeholders. It would be advisable to briefly explain this theory, or at least the meaning and relevance of “stakeholders,” according to it.  The references put down (13, 14) are not treatises on this theory.

AR: We apologize for the imperfections of the first revision. In the latest revised manuscript, we further added to the definition and explanation of stakeholder theory in the introduction (Page 1, section 1, lines 83-85). The following are the contents of the additions based on this comment.

1. Introduction

   The stakeholder theory identifies the generation of value as a central driver of the enterprise, this value is to be shared by a group of stakeholders who can affect or be affected by an organization [15].

  • Comments #4

RC: The change for Figure 1 makes it much more understandable. Table 1 for the hypotheses seems to contribute to keep them in mind and explain the variables, as well.

AR: We thank you for previous valid advice which increases the readability of the paper.

  • Comments #5

RC: Models 1” (line 411) should read “Model 1.” This is written in blue, which are the corrections or additions. There are more mistakes, though.

AR: We apologize for the mistake and appreciate the reviewer for pointing out this issue. We have revised the mistake in our updated draft (Page 11, section 4.2, lines 413 and 415). In addition, we thoroughly checked the original manuscript and modified similar errors.

4.2      Regression Analysis

   Models 1 examines the direct effects of end-of-pipe technologies and clean technolo-gies on economic performance  

4.2      Regression Analysis

   Models 1 also shows that the coefficient for PATENT is significant and positive  

  • Comments #6

RC: In Tables 3, 4 and 5, there should be an explanation of the parentheses and the relationship with the models. This in order to better the understanding of the information.

AR: We appreciate your careful reading and suggestions. In the revised manuscript, we further added to the relevant content for interpreting parentheses and the relationship with the models (Pages 11-16). The following are the contents of the additions based on this comment.

4.2      Regression Analysis

   Ceteris paribus, the results indicate that if a manufacturing firm's end-of-pipe technologies investment increases by 1% or the number of green patent applications increase by 1, the company will see a 0.159 and 0.0028% increase in ROA respectively.  

   That is, ceteris paribus, if companies apply for an additional green patent, enterprises with average financial constraints level will see a 0.0074% decrease in financial constraints.   

   Ceteris paribus, a 1% increase in end-of-pipe technologies investment reduces ROA by 0.0369.   

4.3.1 Heckman two-stage procedure

   The result from Panel A indicates that manufacturing companies participating in Carbon Emission Trading are 26.71% more likely to adopt environmental technologies compared to non-pilot manufacturing companies  

4.3.2 Accuracy of Independent Variable Measurement

   Ceteris paribus, if companies apply for an additional green patent, enterprises will see a 0.0167% increase in TFP.